# Causal relationship between Helicobacter pylori antibodies and gastroesophageal reflux disease (GERD): A mendelian study

Jiaqin Chen[1,2⊚], Junning Zhang[1,3⊚], Xiaolu Ma[1,2], Yuehan Ren[1], Yi Tang[4], Zhongmian Zhang[1,2], Wangyu Ye[1,2], Xiyan Zhang[1,2], Zili Lin[1,2], Lan Wang[1,2], Zhihong Li[1,2]*

1 Graduate School, Beijing University of Chinese Medicine, Beijing, China, 2 Department of Gastroenterology, Dongzhimen Hospital, Beijing University of Chinese Medicine, Beijing, China, 3 Department of Oncology of Integrative Chinese and Western Medicine, China-Japan Friendship Hospital, Beijing, China, 4 School of Traditional Chinese Medicine, Beijing University of Chinese Medicine, Beijing, China

⊚ These authors contributed equally to this work.
* lzhhsm@126.com

**Data Availability Statement:** All files are available from the GWAS database (https://www.ebi.ac.uk/gwas/)(accession numbers: ukb-b-19886, ieu-b-4905, ebi-a-GCST90006913, ebi-a-

## Abstract

### Background

Observational studies have indicated that both Helicobacter pylori infection and the presence of Helicobacter pylori antibodies may increase the risk of gastroesophageal reflux disease (GERD). However, the exact association between Helicobacter pylori antibodies and the occurrence of GERD remains largely unresolved. Therefore, this two-sample Mendelian randomization (MR) study aims to investigate the causal relationship between Helicobacter pylori infection and GERD.

### Methods

This study encompassed seven different specific protein antibodies targeting Helicobacter pylori and utilized a genome-wide association study (GWAS) on GERD. MR analysis was conducted to assess the causal relationship between Helicobacter pylori antibodies and the development of GERD.

### Results

Genetically predicted serum levels of Helicobacter pylori IgG antibodies were positively associated with an increased risk of GERD (odds ratio [OR] = 1.001, 95% CI 1.000–1.003, P = 0.043). No causal relationship was found between other Helicobacter pylori antibodies and gastroesophageal reflux disease.

### Conclusion

The outcomes derived from our two-sample Mendelian randomization analysis demonstrate a discernible link between the levels of Helicobacter pylori IgG antibodies and an augmented

GCST90006914, ebi-a-GCST90006915, ebi-a-GCST90006916, ebi-a-GCST90006912, ebi-a-GCST90006911).

**Funding:** Transverse project of Dongzhimen Hospital of Beijing University of Traditional Chinese Medicine: Improvement effect of Lactobacillus plantarum HCS-001 on gastric discomfort symptoms in Helicobacter pylori infected patients (No. HX-DZM-202269). The funders had no role in study design, data collection and analysis, decision to publish, or preparation of the manuscript.

**Competing interests:** The authors have declared that no competing interests exist.

susceptibility to GERD. However, it is imperative to expand the sample size further in order to corroborate the correlation between Helicobacter pylori infection and GERD.

## Introduction

Gastroesophageal reflux disease (GERD) is a prevalent gastrointestinal disorder frequently encountered in clinical settings, particularly in Western countries. It affects approximately 20% of the population and nearly 40% of children [1], making it a prevalent and widespread condition. GERD is known to cause distressing symptoms that significantly impact individuals' quality of life [2], GERD significantly compromises patients' quality of life and imposes a substantial economic burden [3], The clinical manifestations of GERD commonly include a burning sensation behind the sternum, characterized by the regurgitation of gastric contents into the throat, leading to mucosal damage of the affected organs [4], three phenotypic presentations of GERD exist. Including non-erosive reflux disease (NERD), erosive esophagitis (EE) and BE [5]. BE represents a precancerous condition of the esophagus preceding the development of esophageal cancer [6].

Helicobacter pylori is a Gram-negative bacterium. It is widely believed that Helicobacter pylori infection impacts a substantial proportion of the global population, with a prevalence surpassing 60.3%. Economically disadvantaged regions experience even higher rates, exceeding 80% in some cases [7], Helicobacter pylori plays a vital role in gastric mucosal inflammation and the carcinogenesis process. The potential causal relationship between Helicobacter pylori infection and the occurrence of GERD is currently a topic of debate. Several studies suggest that Helicobacter pylori infection might be a contributing factor in the development of esophageal diseases. Further research is needed to fully understand the complex interaction between Helicobacter pylori and GERD [8]. A study involving 156 patients with peptic ulcers and reflux esophagitis found that eradication of Helicobacter pylori led to a significant improvement in gastroesophageal reflux symptoms compared to patients with persistent infection [9], however, there have been studies that arrived at the opposite conclusion, suggesting that Helicobacter pylori infection may actually serve as a protective factor for GERD [10]. It is important to note that all the evidence currently available is based on observational studies, which inherently have notable limitations and are susceptible to confounding factors, reverse causation, or other biases due to unmeasured or inaccurately measured variables. The causal association between Helicobacter pylori infection and GERD is still uncertain, as there is a dearth of conclusive evidence in this regard. Our study is essential to gain a clearer understanding of the connection between Helicobacter pylori and GERD.

Mendelian randomization (MR) is a robust method that utilizes genetic variations to assess whether the observed correlations between risk factors and outcomes are indicative of causal effects. In the current investigation, a two-sample MR analysis was employed to predict the causal link between Helicobacter pylori infection and GERD, the aim of this study is to investigate whether current or past Helicobacter pylori infection increases the risk of developing GERD, and to assess the potential of Helicobacter pylori eradication as a preventive or therapeutic approach for GERD. The findings of this study will provide valuable recommendations for clinical applications and treatment strategies.

## Study design

MR Analysis is a type of instrumental variable analysis that uses genetic variants as instrumental variables. This methodology capitalizes on the inherent randomness and natural assortment of genetic variations during meiosis, resulting in a randomized dispersion of genetic variants across the population [11], Genetic variation serves as an instrumental variable for the exposure, as it is randomly allocated during conception, thus minimizing residual confounding to the greatest extent possible [12]. The MR design offers a rigorous approach to elucidate causal relationships between complex diseases. The availability of genome-wide association study (GWAS) data has greatly enhanced the accessibility and widespread utilization of two-sample MR designs. In the current investigation, a univariable Mendelian randomization analysis was utilized to explore the causal influence of Helicobacter pylori infection on GERD and unveil plausible underlying mechanisms. The schematic representation of the study design can be found in Fig 1. The data used for this analysis were sourced from previously published GWAS. It is important to note that all original studies obtained ethical consent. Helicobacter pylori infection was defined based on the measurement of serum-specific antibodies targeting Helicobacter pylori proteins, incorporating data from seven different antibody measurements. The resultant dataset comprised 16,404 individuals from the IEU GWAS database, including levels of IgG, Cag A, Vac A, UREA, hydrogen peroxide enzyme, OMP, and GroEL antibodies against Helicobacter pylori. These data can be found on the IEU OpenGWAS project (mrcieu.ac.uk) and all study participants were of European ancestry, both men and women were included. GERD candidate genes were extracted from the IEU GWAS database for this study, including 4,360 cases and 458,650 controls. MR analysis relies on three critical assumptions [13]. Assumption one posits that the instrumental variable, represented by genetic variation in this scenario, ought to exhibit a robust association with the focal exposure. Assumption two posits that the genetic variation should not be associated with any confounding factors that could potentially impact both the exposure and the outcome. Assumption three implies that the genetic variation affects the outcome exclusively through its direct influence on the exposure.

In this MR study, we employed single nucleotide polymorphisms (SNPs) obtained from GWAS as instrumental variables (IVs). Initially, we employed a p-value threshold of 5E-06 for IV selection, aiming to obtain a sufficient number of IVs and bolster statistical power. It is worth noting that this threshold has been commonly employed in comparable studies [14], furthermore, we conducted linkage disequilibrium (LD) clumping by aggregating genetic variants By applying a threshold of $R^2 < 0.001$ within a window size of 10,000 kb. To address potential horizontal pleiotropy, we manually selected proxies from the Pheno Scanner database (http://phenoscanner.medschl.cam.ac.uk) and assessed their association with any previously identified confounding traits ($p < 5E-08$). IVs associated with GERD were excluded. To address the issue of weak instrument bias and its influence on causal inference, we assessed the strength of instrumental variables (IVs) using the formula $F = b^2 exposure / SE^2 exposure$ [15], SNPs with an F-statistic below 10 were deemed weak instrumental variables (IVs) and were excluded from the analysis [16]. We assessed the causal effects of GERD outcomes using five distinct approaches: weighted median, MR-Egger regression, inverse-variance weighted (IVW), weighted mode, and simple mode. The primary statistical model utilized in this investigation was the random-effects inverse-variance weighted (IVW) method, which effectively addresses heterogeneity [17], the IVW method combines the inverse of each IV's variance as weights, disregarding the intercept term in the regression, to yield a weighted average of the IV effects as the final outcome. In contrast, MR-Egger regression takes into account the inclusion of the intercept term in the regression and employs the inverse of the residual variance as weights for fitting. The weighted median estimator (WME) represents the median value

**Exposure**
H. pylori antibodies
(IgG, CagA, VacA, UreA, Catalase, GroEL, OMP)

↓

**Data Preparetion**
P<5E-06
r²<0.001,clumping window=10000kb
Remove IVs related to risk factors of outcome

↓

**MR analysis**
Inverse-variance weighted model(IVW), MR-Egger,
weighted median, simple mode, weighted mode

↓

**Sensitivity Analysis**
Cochrane's Q test, MR-Egger intercept test,
MR-PRESSO test, leave-one-out analysis

**Fig 1. Flow chart.** Schematic diagram of the MR study of the causal relationship between Helicobacter pylori infection and gastroesophageal reflux disease.

derived from the weighted empirical density function of the ratio estimate in question. which represents the consistently estimated causal relationship by half of the valid instrumental variables utilized in the analysis.

## Sensitivity analysis

Due to potential heterogeneity arising from different data sources, MR analysis may introduce bias in the estimation of causal effects. An extensive sensitivity analysis was conducted to assess the robustness and reliability of the MR results, ensuring the validity of the findings in the face of potential confounding factors. Within this study, the primary instrumental variable weighted (IVW) and Mendelian randomization Egger (MR-Egger) methods were evaluated for heterogeneity. The presence of heterogeneity was evaluated using Cochran's Q test, a widely utilized statistical method in medical research. This test allowed for the assessment and quantification of potential variations and discrepancies among the analyzed data, contributing to a comprehensive understanding of the study's findings. To assess the credibility of the causal

estimates, the sensitivity analysis encompassed a range of methodologies in the present study. These included the utilization of the MR-Egger intercept test, and leave-one-out analysis. By incorporating these rigorous approaches, the research aimed to comprehensively evaluate and validate the robustness of the causal inferences derived from the MR framework. This comprehensive sensitivity analysis bolstered the reliability and confidence in the study's findings, enhancing its contribution to the field of medicine. Statistical analyses and data visualizations were performed utilizing R version 4.2.1. Mendelian randomization analyses were performed with the use of the Two Sample MR Package, version 0.5.6.

## Result

Our MR analysis demonstrated a causal link between genetically predicted levels of Helicobacter pylori IgG antibodies and the occurrence of GERD, although the findings were not entirely consistent. Using five different methods to analyze the antibody levels of Helicobacter pylori IgG, CagA, VacA, UREA, hydrogen peroxide enzyme, OMP, and GroEL, under the IVW method, an increased risk of GERD was associated with Helicobacter pylori IgG antibody levels (odds ratio [OR] = 1.003, 95% CI 1.00–1.004, P = 0.043) (See Fig 2). In order to address the potential influence of pleiotropy, one important tests were performed: the MR-Egger intercept test. The MR-Egger analysis, a statistical method designed to detect potential horizontal pleiotropy, was conducted to examine whether there were any genetic variants influencing both the exposure (Helicobacter pylori infection) and the outcome (GERD) through pathways unrelated to the causal relationship of interest. Remarkably, the results of the MR-Egger intercept test exhibited no significant evidence of horizontal pleiotropy, as indicated by all P-values surpassing the threshold of 0.05. This outcome suggests that the instrumental variables employed in the MR analysis were not biased by horizontal pleiotropy, reinforcing the credibility of the causal inference made regarding the association between Helicobacter pylori infection and GERD risk. Cochran's Q test and $I^2$ statistics indicated low heterogeneity and high reliability of these SNPs (all P > 0.05) (See Table 1). The funnel plot exhibited general symmetry, suggesting limited evidence of heterogeneity. (Refer to the Supplement materials)

## Discussion

The association between Helicobacter pylori infection and GERD continues to be a subject of ongoing debate and scientific inquiry, and currently, there is no definitive causal relationship established. Although previous observational studies have been limited, the complexity of this relationship arises from various factors, including the heterogeneity of study populations, differences in study designs, and the multifactorial nature of GERD etiology, several studies [18–22] have found an increased risk of GERD in both Eastern and Western populations with Helicobacter pylori infection, in a Japanese population, Helicobacter pylori infection was significantly associated with short-segment Barrett's esophagus(SSBE), it may be a risk factor for SSBE [18]. At the same time, three studies conducted in Lithuania [19], Hong Kong [20], and the United States [22] respectively demonstrated that Helicobacter pylori is a statistically significant predictive factor for NERD. A study [21] conducted in Sweden revealed that adults infected with Helicobacter pylori (HP) in two cities in Sweden have a higher risk of developing symptoms of gastroesophageal reflux disease. An Iranian study showed that Helicobacter pylori infection is a significant and correlated risk factor for esophagitis [23], furthermore, scientific research has established a correlation between Helicobacter pylori infection and mild GERD. These findings underscore the association between Helicobacter pylori infection and the presence of mild GERD symptoms [24], in a randomized controlled study, it was observed that individuals with Helicobacter pylori infection exhibited a higher prevalence of heartburn

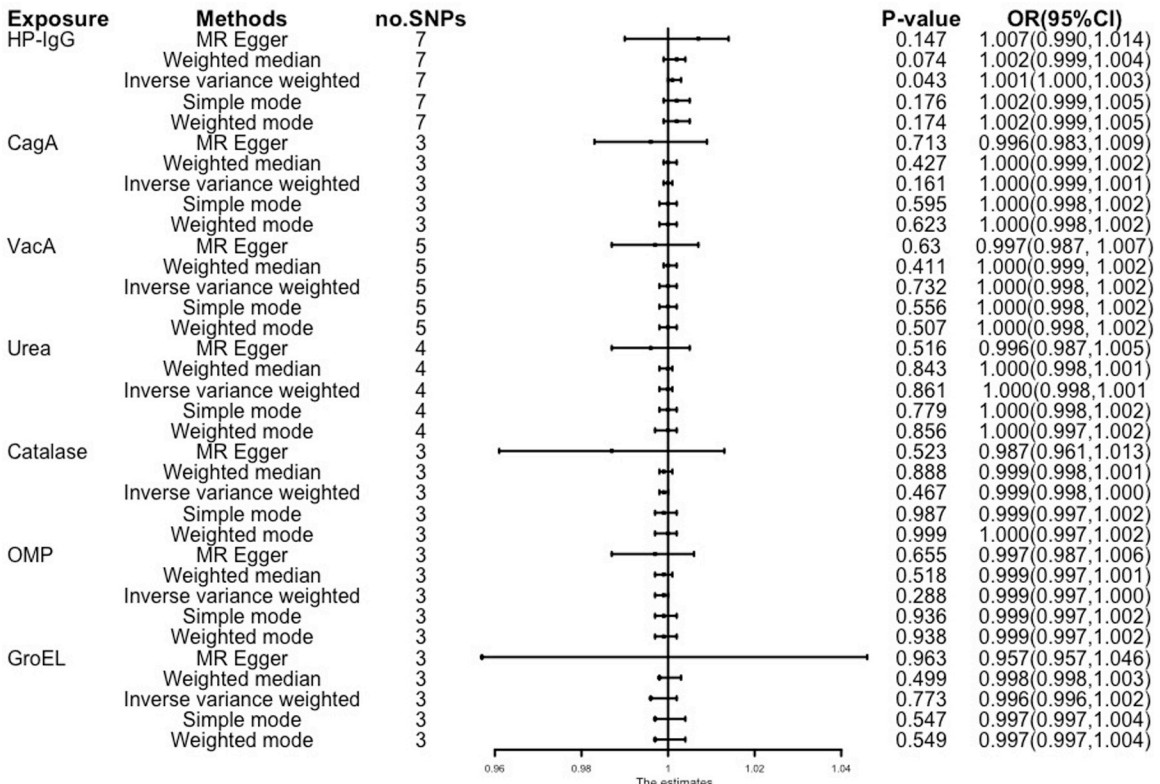

| Exposure | Methods | no.SNPs | P-value | OR(95%CI) |
|---|---|---|---|---|
| HP-IgG | MR Egger | 7 | 0.147 | 1.007(0.990,1.014) |
| | Weighted median | 7 | 0.074 | 1.002(0.999,1.004) |
| | Inverse variance weighted | 7 | 0.043 | 1.001(1.000,1.003) |
| | Simple mode | 7 | 0.176 | 1.002(0.999,1.005) |
| | Weighted mode | 7 | 0.174 | 1.002(0.999,1.005) |
| CagA | MR Egger | 3 | 0.713 | 0.996(0.983,1.009) |
| | Weighted median | 3 | 0.427 | 1.000(0.999,1.002) |
| | Inverse variance weighted | 3 | 0.161 | 1.000(0.999,1.001) |
| | Simple mode | 3 | 0.595 | 1.000(0.998,1.002) |
| | Weighted mode | 3 | 0.623 | 1.000(0.998,1.002) |
| VacA | MR Egger | 5 | 0.63 | 0.997(0.987, 1.007) |
| | Weighted median | 5 | 0.411 | 1.000(0.999, 1.002) |
| | Inverse variance weighted | 5 | 0.732 | 1.000(0.998, 1.002) |
| | Simple mode | 5 | 0.556 | 1.000(0.998, 1.002) |
| | Weighted mode | 5 | 0.507 | 1.000(0.998, 1.002) |
| Urea | MR Egger | 4 | 0.516 | 0.996(0.987,1.005) |
| | Weighted median | 4 | 0.843 | 1.000(0.998,1.001) |
| | Inverse variance weighted | 4 | 0.861 | 1.000(0.998,1.001 |
| | Simple mode | 4 | 0.779 | 1.000(0.998,1.002) |
| | Weighted mode | 4 | 0.856 | 1.000(0.997,1.002) |
| Catalase | MR Egger | 3 | 0.523 | 0.987(0.961,1.013) |
| | Weighted median | 3 | 0.888 | 0.999(0.998,1.001) |
| | Inverse variance weighted | 3 | 0.467 | 0.999(0.998,1.000) |
| | Simple mode | 3 | 0.987 | 0.999(0.997,1.002) |
| | Weighted mode | 3 | 0.999 | 1.000(0.997,1.002) |
| OMP | MR Egger | 3 | 0.655 | 0.997(0.987,1.006) |
| | Weighted median | 3 | 0.518 | 0.999(0.997,1.001) |
| | Inverse variance weighted | 3 | 0.288 | 0.999(0.997,1.000) |
| | Simple mode | 3 | 0.936 | 0.999(0.997,1.002) |
| | Weighted mode | 3 | 0.938 | 0.999(0.997,1.002) |
| GroEL | MR Egger | 3 | 0.963 | 0.957(0.957,1.046) |
| | Weighted median | 3 | 0.499 | 0.998(0.998,1.003) |
| | Inverse variance weighted | 3 | 0.773 | 0.996(0.996,1.002) |
| | Simple mode | 3 | 0.547 | 0.997(0.997,1.004) |
| | Weighted mode | 3 | 0.549 | 0.997(0.997,1.004) |

**Fig 2. Forest plot.** Mendelian randomization results of the effect of Helicobacter pylori infection on clinical traits associated with gastroesophageal reflux disease.

symptoms in comparison to those without Helicobacter pylori infection [25], The findings of this study indicate an elevated likelihood of experiencing symptoms of heartburn in association with Helicobacter pylori infection. The presence of Helicobacter pylori infection appears to be associated with an increased susceptibility to developing or experiencing symptoms of heartburn. The malfunctioning of the lower esophageal sphincter is considered the primary contributor to the development of GERD. Simultaneously, the gastric mucosa produces a variety of cytokines which activate immune cell recruitment and migration and contribute to the development of the disease [26]. Multiple theoretical mechanisms are implicated in the development of GERD caused by Helicobacter pylori infection. Certain studies propose that Helicobacter pylori infection can impact the secretion of gastric acid [25], Considering that Helicobacter pylori infection primarily elicits gastritis, it leads to increased gastric acid secretion, resulting in increased acid exposure to the lower esophagus, leading to symptoms such as acid reflux and heartburn. Additional research has revealed that the existence of Helicobacter pylori hampers the mucosal healing capacity among individuals with reflux esophagitis, thereby intensifying symptoms of reflux [27]. Moreover, prospective studies have unequivocally revealed a compelling association between Helicobacter pylori infection and a notable escalation in inflammatory processes transpiring at the delicate juncture between the esophagus and the stomach among patients afflicted with GERD [28]. Our MR study supports a direct relationship between Helicobacter pylori infection and GERD, MR Study enable the derivation of more robust and evidence-based conclusions, offering advantages over other methods without these issues. Our study is the first to reveal a causal relationship between H. pylori infection

**Table 1. Causal relationship between the Helicobacter pylori antibodies and GERD.**

| Exposure | Methods | N SNPs | MR analysis | | Heterogeneity | | MR-Egger regression | |
|---|---|---|---|---|---|---|---|---|
| | | | OR(95%CI) | P-value | Cochran's Q | P value | Egger intercept | P value |
| IgG | MR Egger | 7 | 1.007(0.990,1.014) | 0.147 | 1.819390 | 0.610 | -0.0008 | 0.224 |
| | Weighted median | 7 | 1.002(0.999,1.004) | 0.074 | | | | |
| | Inverse variance weighted | 7 | 1.001(1.000,1.003) | 0.043 | 4.146256 | 0.386 | | |
| | Simple mode | 7 | 1.002(0.999,1.005) | 0.176 | | | | |
| | Weighted mode | 7 | 1.002(0.999,1.005) | 0.174 | | | | |
| CagA | MR Egger | 3 | 0.996(0.983,1.009) | 0.713 | 0.1791725 | 0.672 | 0.0009 | 0.651 |
| | Weighted median | 3 | 1.000(0.999,1.002) | 0.427 | | | | |
| | Inverse variance weighted | 3 | 1.000(0.999,1.001) | 0.161 | 0.5510087 | 0.759 | | |
| | Simple mode | 3 | 1.000(0.998,1.002) | 0.595 | | | | |
| | Weighted mode | 3 | 1.000(0.998,1.002) | 0.623 | | | | |
| VacA | MR Egger | 5 | 0.997(0.987, 1.007) | 0.63 | 9.856508 | 0.019 | 0.0006 | 0.589 |
| | Weighted median | 5 | 1.000(0.999, 1.002) | 0.411 | | | | |
| | Inverse variance weighted | 5 | 1.000(0.998, 1.002) | 0.732 | 11.048687 | 0.026 | | |
| | Simple mode | 5 | 1.000(0.998, 1.002) | 0.556 | | | | |
| | Weighted mode | 5 | 1.000(0.998, 1.002) | 0.507 | | | | |
| Urea | MR Egger | 4 | 0.996(0.987,1.005) | 0.516 | 1.143195 | 0.564 | 0.0006 | 0.496 |
| | Weighted median | 4 | 1.000(0.998,1.001) | 0.843 | | | | |
| | Inverse variance weighted | 4 | 1.000(0.998,1.001 | 0.861 | 1.810789 | 0.612 | | |
| | Simple mode | 4 | 1.000(0.998,1.002) | 0.779 | | | | |
| | Weighted mode | 4 | 1.000(0.997,1.002) | 0.856 | | | | |
| Catalase | MR Egger | 3 | 0.987(0.961,1.013) | 0.523 | 0.2561435 | 0.612 | 0.002 | 0.536 |
| | Weighted median | 3 | 0.999(0.998,1.001) | 0.888 | | | | |
| | Inverse variance weighted | 3 | 0.999(0.998,1.000) | 0.467 | 1.0509169 | 0.591 | | |
| | Simple mode | 3 | 0.999(0.997,1.002) | 0.987 | | | | |
| | Weighted mode | 3 | 1.000(0.997,1.002) | 0.999 | | | | |
| OMP | MR Egger | 3 | 0.997(0.987,1.006) | 0.655 | 0.8282778 | 0.363 | 0.0004 | 0.744 |
| | Weighted median | 3 | 0.999(0.997,1.001) | 0.518 | | | | |
| | Inverse variance weighted | 3 | 0.999(0.997,1.000) | 0.288 | 1.0087270 | 0.604 | | |
| | Simple mode | 3 | 0.999(0.997,1.002) | 0.936 | | | | |
| | Weighted mode | 3 | 0.999(0.997,1.002) | 0.938 | | | | |
| GroEL | MR Egger | 3 | 0.957(0.957,1.046) | 0.963 | 5.168425 | 0.023 | 0.0030 | 0.951 |
| | Weighted median | 3 | 0.998(0.998,1.003) | 0.499 | | | | |
| | Inverse variance weighted | 3 | 0.996(0.996,1.002) | 0.773 | 5.199087 | 0.074 | | |
| | Simple mode | 3 | 0.997(0.997,1.004) | 0.547 | | | | |
| | Weighted mode | 3 | 0.997(0.997,1.004) | 0.549 | | | | |

The impact of Helicobacter pylori infection on the clinical characteristics linked to acid reflux disease was examined using Mendelian randomization.

and GERD, and this study can improve the understanding of the pathogenic factors of GERD from a systems biology perspective.

This study has several limitations. Firstly, the sample size utilized in the GWAS pertaining to Helicobacter pylori infection was relatively modest in scale, which may introduce some bias and warrants further investigation with larger sample sizes. Secondly, a p-value threshold of 5E-06 is employed for instrumental variable selection, with the potential to introduce a certain degree of weak instrument bias that can impact the overall estimation. Thirdly, the scope of

our study was confined to individuals with European ancestry, thereby potentially imposing constraints on the applicability of our findings to broader populations.

## Conclusion

Based on the comprehensive analysis of our study data, compelling evidence emerges to support a robust causal nexus between the genetically predicted levels of Helicobacter pylori IgG antibodies and a discernible augmentation in the vulnerability to GERD. However, it is imperative to underscore that additional endeavors encompassing more expansive cohorts are imperative to corroborate and fortify the intricate interplay between Helicobacter pylori infection and the manifestation of GERD. The pursuit of future research endeavors, characterized by enlarged sample sizes, shall undoubtedly engender a deeper understanding of the intricate pathophysiological mechanisms underpinning this intriguing association.

## Supporting information

**S1 Table. Mendelian randomization analysis of Hp and GERD.** Instrumental SNPs of GERD.
(DOCX)

**S1 Fig. Scatter plot.** (A) Hp-CagA and GERD; (B) Hp-Catalase and GERD; (C) Hp-GroEL and GERD; (D) Hp-IgG and GERD; (E) Hp-OMP and GERD; (F) Hp-UREA and GERD; (G) Hp-VacA and GERD.
(DOCX)

**S2 Fig. Funnel plot.** (A) Hp-CagA and GERD; (B) Hp-Catalase and GERD; (C) Hp-GroEL and GERD; (D) Hp-IgG and GERD; (E) Hp-OMP and GERD; (F) Hp-UREA and GERD; (G) Hp-VacA and GERD.
(DOCX)

**S3 Fig. Leave-one-out sensitivity analysis.** (A) Hp-CagA and GERD; (B) Hp-Catalase and GERD; (C) Hp-GroEL and GERD; (D) Hp-IgG and GERD; (E) Hp-OMP and GERD; (F) Hp-UREA and GERD; (G) Hp-VacA and GERD.
(DOCX)

**S4 Fig. Forest plot.** (A) Hp-CagA and GERD; (B) Hp-Catalase and GERD; (C) Hp-GroEL and GERD; (D) Hp-IgG and GERD; (E) Hp-OMP and GERD; (F) Hp-UREA and GERD; (G) Hp-VacA and GERD.
(DOCX)

## Acknowledgments

We would like to thank all participants and investigators who contributed to the GWAS data.

## Author Contributions

**Formal analysis:** Junning Zhang.

**Writing – original draft:** Jiaqin Chen.

**Writing – review & editing:** Xiaolu Ma, Yuehan Ren, Yi Tang, Zhongmian Zhang, Wangyu Ye, Xiyan Zhang, Zili Lin, Lan Wang, Zhihong Li.

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
