## [Decision Letter · Decision Letter 0]

16 Oct 2023

PONE-D-23-28111Causal Relationship Between Helicobacter pylori Antibodies and gastroesophageal reflux disease (GERD): A Mendelian StudyPLOS ONE

Dear Dr. Li,

Thank you for submitting your manuscript to PLOS ONE. After careful consideration, we feel that it has merit but does not fully meet PLOS ONE’s publication criteria as it currently stands. Therefore, we invite you to submit a revised version of the manuscript that addresses the points raised during the review process.

We look forward to receiving your revised manuscript.

Kind regards,

Yasin Sahin

Academic Editor

PLOS ONE

Journal Requirements:

Transverse project of Dongzhimen Hospital of Beijing University of Traditional Chinese Medicine: Improvement effect of Lactobacillus plantarum HCS-001 on gastric discomfort symptoms in Helicobacter pylori infected patients (No. HX-DZM-202269).

None declared.

6. Please include captions for your Supporting Information files at the end of your manuscript, and update any in-text citations to match accordingly. Please see our Supporting Information guidelines for more information: http://journals.plos.org/plosone/s/supporting-information

Additional Editor Comments:

Thanks to the authors for the study.

I invite you to resubmit your manuscript after addressing two reviewers’ comments. When resubmitting your manuscript, please carefully consider all issues mentioned in the reviewers' comments, outline every change made point by point, and provide suitable rebuttals for any comments not addressed.

Reviewers' comments:

Reviewer's Responses to Questions

**Comments to the Author**

1. Is the manuscript technically sound, and do the data support the conclusions?

Reviewer #1: Yes

Reviewer #2: Yes

2. Has the statistical analysis been performed appropriately and rigorously? 

Reviewer #1: N/A

Reviewer #2: No

3. Have the authors made all data underlying the findings in their manuscript fully available?

Reviewer #1: Yes

Reviewer #2: No

4. Is the manuscript presented in an intelligible fashion and written in standard English?

Reviewer #1: Yes

Reviewer #2: Yes

5. Review Comments to the Author

Reviewer #1: Dear Editor,

I should first thank for inviting me as potential reviewer to read and comment on paper entitled ‘’Causal Relationship Between Helicobacter pylori Antibodies and gastroesophageal reflux disease (GERD): A Mendelian Study’’.

In the current study, the authors aimed to investigate the causal relationship between Helicobacter pylori infection and GERD.

The main title accurately reflects the major topic and content of the study.

The abstract summarizes and reflects the work described in the manuscript. Also, the abstract presents the significant points related to the background, objectives, materials and methods, results and conclusions. The materials and methods sufficiently described for the results and conclusions that are presented in the preceding sections. The study type and design were defined in the section of the materials and methods. Figures are sufficient and well organized.

The section of the discussion is well organized. The conclusions are drawn appropriately supported by the literature. The manuscript adequately describes the background, present status and significance of the study. The manuscript interprets the findings adequately and appropriately, highlighting the key points clearly.

I think that it will contribute to the literature. I have some minor criticisms.

- All of the manuscript should be checked n terms of spelling errors.

- The manuscript appropriately cites the important and authoritative references but does not cite the recent published articles about Helicobacter pylori. If the recent published article about celiac disease for example ‘’Relationship between the severity of Helicobacter pylori infection and neutrophil and lymphocyte ratio and mean platelet volume in children. Arch Argent Pediatr 2020 Jun; 118(3):e241-e245’’ are cited, the manuscript would be better. The authors can benefit from this recent study.

Reviewer #2: Some revisions are required. ıntroduction was written so long. ıt should be shortened. I could not find ethical approval number and its year. some statistical informations should be added. In addition the finding of the study was not compared adequately with similar studies.

6. PLOS authors have the option to publish the peer review history of their article (what does this mean?). If published, this will include your full peer review and any attached files.

Reviewer #1: No

Reviewer #2: No

---

## [Author Response · Author response to Decision Letter 0]

5 Nov 2023

Dear Editor and Reviewers, 

Thank you for giving us the opportunity to revise our manuscript "Causal Relationship Between Helicobacter pylori Antibodies and gastroesophageal reflux disease (GERD): A Mendelian Study" (Submission ID PONE-D-23-28111). We gratefully appreciate the valuable comments and suggestions provided by the reviewers, and we believe that their input has greatly improved our manuscript. 

We have carefully considered the reviewers’ suggestions and have made comments and revisions to address all their concerns. Our responses to the editor’s and reviewers’ comments are in "Response to Reviewers" (marked with ‘>>Reply:’ and highlighted in blue). In the revised manuscript, red highlighted text indicates where changes have been made. We have responded to all the issues identified in document “PONE-D-23-28111_reviewer-REVISION”, as detailed in “PONE-D-23-28111_author-REVISION”.

We hope this revised version is acceptable for publication.

On behalf of all authors,

Jiaqin Chen & Junning Zhang

Corresponding Author:

Zhihong Li*

Email Address: lzhhsm@126.com

MD., Ph.D.; Professor

Department of Gastroenterology, 

Dongzhimen Hospital, Beijing University of Chinese Medicine. Beijing, China.

---

## [Editor Report · Decision Letter 1]

9 Nov 2023

Causal Relationship Between Helicobacter pylori Antibodies and gastroesophageal reflux disease (GERD): A Mendelian Study

PONE-D-23-28111R1

Dear Dr. Zhihong Li,

We’re pleased to inform you that your manuscript has been judged scientifically suitable for publication and will be formally accepted for publication once it meets all outstanding technical requirements.

Kind regards,

Yasin Sahin

Academic Editor

PLOS ONE

Additional Editor Comments (optional):

Thanks to the authors for the study.

I think that it will contribute to the literature.
---

## [Editor Report · Acceptance letter]

27 Nov 2023

PONE-D-23-28111R1 

Causal Relationship Between Helicobacter pylori Antibodies and gastroesophageal reflux disease (GERD): A Mendelian Study 

Dear Dr. Li:

I'm pleased to inform you that your manuscript has been deemed suitable for publication in PLOS ONE. Congratulations! Your manuscript is now with our production department. 

Kind regards, 

on behalf of

Dr. Yasin Sahin 

Academic Editor

PLOS ONE